# Targeted psychological and psychosocial interventions for auditory hallucinations in persons with psychotic disorders: Protocol for a systematic review and meta-analysis

Laura Fässler[1,2]*, Irene Bighelli[3,4], Stefan Leucht[3,4], Michel Sabé[5], Malek Bajbouj[1,2,4], Christine Knaevelsrud[2,4], Kerem Böge[1,4,6]

1 Department of Psychiatry and Neurosciences, Charité–University Medicine Berlin, Berlin, Germany, 2 Department of Education and Psychology, Freie Universität Berlin, Berlin, Germany, 3 Department of Psychiatry and Psychotherapy, Technical University of Munich, School of Medicine and Health, Munich, Germany, 4 German Center for Mental Health (DZPG), Berlin, Germany, 5 Division of Psychiatric Specialties, Department of Psychiatry, University Hospitals of Geneva, Geneva, Switzerland, 6 Medical University Brandenburg–Theodor Fontane, Neuruppin, Germany

* laura.faessler@charite.de

**Data Availability Statement:** No datasets were generated or analysed during the current study

## Abstract

### Background

In recent years, a growing body of evidence has demonstrated the efficacy of non-pharmacological interventions for schizophrenia spectrum disorders (SSD) including positive symptoms such as auditory hallucinations (AH). However, clinical trials predominantly examine general treatment effects for positive symptoms. Therefore, previous research is lacking in comprehensive and clear evidence about psychological and psychosocial approaches that are primarily tailored to treat AH. To overcome this knowledge gap in the current literature, we will conduct a systematic review and meta-analysis to assess the efficacy of clearly targeted psychological and psychosocial interventions for AH in persons with SSD.

### Methods and analysis

This study protocol has been developed according to the guidelines of the Preferred Reporting Items for Systematic Reviews and Meta-Analysis Protocols. We will include all randomized controlled trials analyzing the efficacy of targeted psychological and psychosocial interventions especially aimed at treating AH in SSD. We will include studies on adult patients with SSD experiencing AH. The primary outcome will be the change on a published rating scale measuring AH. Secondary outcomes will be delusions, overall symptoms, negative symptoms, depression, social functioning, quality of life, and acceptability (drop-out). We will search relevant databases and the reference lists of included literature. The study selection process will be conducted by two independent reviewers. We will conduct a random-effect meta-analysis to consider heterogeneity across studies. Analyses will be carried out by software packages in R. The risk of bias in each study will be evaluated using the

protocol. We will publish the R script and all relevant data sets through the openly accessible platform OSF when the planned study is completed and published.

**Funding:** Laura Fässler, L.F. is funded by the Elsa-Neumann scholarship of the federal state of Berlin (doctoral funding). The funders had no role in study design, data collection and analysis, decision to publish, or preparation of the manuscript. Website: https://www.fu-berlin.de/en/sites/drs/funding/nafoeg/index.html Kerem Böge, K.B. is funded by the Medical Scientist Program of the Charité – Universitätsmedizin Berlin, Nachwuchskommission. The funders had no role in study design, data collection and analysis, decision to publish, or preparation of the manuscript. Website: https://nachwuchs.charite.de/postdocs/medical_scientist_pilotprogramm/.

**Competing interests:** The authors have declared that no competing interests exist.

Cochrane Risk of Bias tool. Assessment of heterogeneity and sensitivity analysis will be conducted.

## Discussion

The proposed study will augment the existing evidence by providing an overview of effective treatment approaches and their overall efficacy at treating AH in SSD. These findings will complement existing evidence that may impact future treatment implementations in clinical practice by addressing effective strategies to treat AH and therefore improve outcomes for the addressed population.

## Ethics and dissemination

No ethical issues are foreseen. We will publish the results from this study in peer-reviewed journals and at relevant scientific conferences.

## Trial registration

**PROSPERO registration number:** CRD42023475704.

## Introduction

### Rationale

According to recent changes in the *International Classification of Diseases* [ICD-11; 1], schizophrenia and other primary psychotic disorders are defined by a wide range of characteristic features. Core symptoms include (1) persisting delusions, (2) persisting hallucinations, (3) formal thought disorders, (4) experiences of interferences, passivity, or external surveillance, (5) negative symptoms such as diminished emotions, social withdrawal, or anhedonia, (6) disorganized behavior, and (7) psychomotor symptoms. Besides, cognitive deficits are experienced through deficiency in processing speed and episodic memory [2]. The lifetime prevalence of schizophrenia has been estimated at approximately 1% [3]. Persons with schizophrenia spectrum disorders (SSD, ICD-10 F2.x), including schizophrenia and other primary psychotic disorders, are frequently affected by long-term psychiatric impairments [4], ranking among the eighth leading cause of disability worldwide in persons aged between 15 and 44 years [5]. The annual treatment expenses for the health care system are tremendous, ranging from 9.63 to 13.52 billion euros in Germany alone [6] while being one of the largest components of health service spending in the European area [7]. Chronic courses of the disorders predominantly cause costs, low medication adherence, and psychiatric comorbidities that contribute to a prolonged duration of inpatient treatment [8].

Sensory perceptions such as auditory hallucinations (AH) are considered hallmark characteristics of SSD [9]. Since AH occur in various clinical and non-clinical populations [10], they can be considered along a continuum from normal to abnormal peculiarities rather than a categorical phenomenon [11]. However, some differences in AH characteristics between non-clinical and clinical populations are specified. The estimated lifetime prevalence of AH in the global population is 7.3% [12]. In contrast, samples of people with SSD report a lifetime prevalence between 64–80% [13], indicating a high vulnerability for the phenomenon. Compared to healthy populations, persons with SSD undergo increased distress, frequency, and duration of AH while also experiencing a more negative emotional valence of content [10]. Reasons for the severe distress associated with AH are lacking control over the appearance, intrusive

cognitions, and threatening contents [14]. Summarizing the current state of research, AH occurs as a fluid experience from ordinary to pathological stages depending on the perceived distress and extension of processes.

International [15] and national [16] guidelines on primary interventions for SSD recommend a multi-professional and integrative approach. The present research states a combination of psychopharmacology in conjunction with psychological interventions consisting of individual cognitive-behavioral psychotherapy (CBT), family interventions [for an overview see 15], and other psychosocial interventions [16]. Even though meta-analytic data on antipsychotic pharmacological treatment estimated overall medium effect sizes [17], there are several limitations associated with antipsychotics, such as disabling side effects, poor treatment adherences, or limited response [18–20]. Concerning the current treatment recommendations, several implementation deficits and structural limitations exist in treating SSD.

In recent years, a growing body of evidence suggests the efficacy of non-pharmacological approaches to treating persons with SSD such as cognitive behavioral therapy, mindfulness and social skills training, or metacognitive strategies [21–24]. Psychosocial and psychological interventions for SSD have been examined in a wide range of randomized controlled trials (RCT), systematic reviews, and meta-analyses [25–28]. Still, cognitive behavioral therapy for psychosis (CBTp) remains the most recommended psychological approach to treat SSD, with small to moderate effectiveness for positive symptoms [standardized mean difference, SMD = 0.29; 27]. However, findings are predominantly general regarding particular symptoms since those interventions are mainly applied to treat many aspects of the disorders that solely enable assertions about an unspecific treatment effect. Therefore, symptom-specific interventions aim to improve treatment in distinct symptomatic categories such as AH. Besides, they can address an amelioration of treatment adherence for psychological approaches by its individual formulation [e.g., 29]. This is especially relevant due to the high non-adherence rates in SSD, mainly for pharmacological interventions [30]. A wide range of tailored approaches for AH focus on elements of CBTp, such as the Cognitive Therapy for Command Hallucinations [CTCH; 31]. Besides, concepts are augmenting different approaches, such as CBTp and Acceptance and Commitment Therapy (ACT), aiming to reduce engagement with harmful command hallucinations [32]. Recently, digital applications such as avatar therapy [33, 34] have been used to create an innovative and patient-centered experience that allows for a dialogic interaction with voices, reclaiming the feeling of control. Despite this, approaches such as relating therapy [35] or mindfulness-based interventions for voice hearing [36, 37] are implemented to address distressing AH. Targeted psychological interventions for AH emerging from CBTp focus on an empathetic and accepting perspective to change the person's relationship with the voices rather than their content or quantity [38–40]. Accordingly, affected persons can increase their perceived control over the appearance of the voices and, therefore, decrease the associated distress [38]. Those interventions based on different therapeutic approaches are primarily intended to treat distressing AH in SSD and voice hearers. Although existing trials indicate separate effect sizes for the individual interventions, no overall efficacy analyses of symptom-specific psychological interventions for AH exist.

Current meta-analytic evidence predominantly examined the efficacy of psychological interventions for SSD, whereby positive symptoms are therapeutically addressed and analyzed simultaneously [e.g., 27, 41, 42]. Furthermore, there are systematic reviews that contribute to aggregate existing evidence regarding psychosocial approaches for SSD, equally in general [23]. However, there is a significant lack in the analysis of tailored interventions and their efficacy in addressing specific symptoms such as AH. Previous research on treatment effects of tailored approaches addressing AH found large effect sizes for experienced focused counselling ($d = 1$) [43], moderate effects for a reduction in the negative impact of voices for individual

mindfulness-based interventions [37], or significant reduction in harmful compliance with and an increase in the perceived control over voices with small-to-moderate effect sizes [31]. To date, there is only one meta-analysis that analyzed the efficacy of individually tailored formulation-based CBTp approaches for AH and delusions [Hedges' g, g (AH) = 0.44, g (delusions) = 0.36; 44]. Compared to recent findings of generic CBTp for positive symptoms with small to moderate effect sizes [27], those results are promising and need further evaluation. Another systematic review summarized especially tailored CBTp interventions for AH and delusions [45]. Equally, the results indicated stronger effects of targeted CBTp approaches for hallucinations than generic CBTp since all studies reported effect sizes above *d* = 0.4 on at least one primary outcome at post-intervention. However, existing reviews are outdated and only focus on CBTp while neglecting other third-wave and digital approaches, such as avatar therapy [44, 45]. To the best of our knowledge, there is no meta-analysis and systematic review focused explicitly and solely on tailored psychological interventions for AH in persons with SSD. Those shortcomings in the present evidence need to be considered in future research to facilitate an overview and proof of the efficacy of those symptom-specific approaches for AH.

### Research aim

To overcome flaws in the knowledge of specialized treatments for AH in SSD, we will conduct a systematic review and meta-analysis of the current evidence of particularly tailored psychological and psychosocial interventions for AH in SSD. We aim to give an overview of available therapies for AH and their overall efficacy. Moreover, the present investigation will contribute to an improved understanding of rather specific than generic effects of psychological approaches for AH. To the best of our knowledge, this is the first meta-analysis that will be conducted on all psychological interventions examined in the framework of RCTs and specially tailored to treat AH in patients with SSD.

### Objectives

To estimate the relative treatment effects of targeted psychological and psychosocial interventions for AH, the following aspects will be addressed in the analysis:

1. Efficacy of the interventions on AH, measured by validated rating scales.

2. Other secondary efficacy measures including delusions, overall symptoms, negative symptoms, depression, social functioning, quality of life, and acceptability measures.

3. Reviewing of currently available and implemented psychosocial and psychological interventions especially tailored for AH in SSD.

The analysis will focus on targeted psychological and psychosocial interventions primarily aimed at treating distressing AH in individuals predominantly diagnosed with SSD. According to previous findings on different psychological and psychosocial interventions for AH and SSD [27, 31, 44, 45], we expect moderate effect sizes for tailored approaches for AH. For a detailed description of the included interventions, see section *Types of Interventions*.

## Methods

### Criteria for study consideration

The proposed methods and structures for this systematic review and meta-analysis have been developed according to the guidelines of the Preferred Reporting Items for Systematic Review and Meta-Analysis Protocols [PRISMA-P; 46]. Furthermore, the study protocol has been

registered on the International Prospective Register of Systematic Reviews (PROSPERO; registration number: CRD42023475704). We will update the record if any amendments or revisions are made.

## Eligibility criteria for studies

All relevant Randomized Controlled Trials (RCTs), including participants with a primary diagnosis of SSD (i.e., schizophrenia, schizoaffective disorder, and other primary psychotic disorders), will be considered. Moreover, participants must receive a psychological and psychosocial intervention as defined below (see section *Types of Interventions*). Open and blinded studies will be included since blinding in interventional trials is mostly only possible for the outcome assessors but not for the providers or participants. The intervention must be compared by a treatment-as-usual (TAU) condition, another intervention, or a wait-list control group. Only research published or translated into English and German will be included. For cross-over studies, we will only analyze the first cross-over phase to avoid complications caused by carry-over effects in psychological treatments. Case series and trials with less than 10 participants in each study arm will be excluded to ensure sensitivity for treatment effects [47], validity, and statistical power of results [48].

## Types of participants

The present review aims to examine the efficacy of specifically tailored psychosocial and psychological interventions for AH in patients with SSD. We will apply the following inclusion and exclusion criteria to select an eligible population. We will only include adult participants aged 18 or older. As stated above, participants diagnosed with SSD (including schizophrenia, schizoaffective disorder, and other primary psychotic disorders) will be included irrespective of the diagnostic criteria applied in each study. This is in line with the strategy of the Cochrane Schizophrenia Group [49] that recommends to not only considering studies that used specific diagnosis systems and diagnostic criteria such as the International Classification of Diseases (ICD; 10th revision) or Diagnostic and Statistical Manual of Mental Disorders (DSM; 5th edition) since those are not consistently and universally used in clinical practice. This approach will help analyze the results in a setting close to clinical routine and increase generalizability.

A primary diagnosis of SSD (including schizophrenia, schizoaffective disorder, and other primary psychotic disorders) is mandatory. In case that studies also included participants with another or no diagnosis, the study needs to consider at least 75% of the sample with a diagnosis of SSD. There will be no restriction on symptom severity or illness duration, wherefore persons with first-episode psychosis will be included. However, participants must have current AH, explicitly mentioned as an inclusion criterion or requirement for participation. This needs to be operationalized and ensured either by ratings scales measuring the frequency of AH with a predefined cut-off score, over outcome measures, self-reports, screening procedures, or clinical documents such as physician's letters. We will exclude studies examining particular subpopulations: (1) Studies in which the examined sample is at risk for psychosis with only prodromal indications of psychosis. (2) Studies focusing on interventions that aim to treat participants with predominantly other positive symptoms, such as delusions or negative symptoms from a population view. (3) Studies that analyzed healthy individuals or children/adolescents under 18 years. (4) Studies that considered AH in other clinical populations with a different primary diagnosis rather than SSD, such as posttraumatic stress disorder, borderline personality disorder or bipolar disorder, or in a non-clinical sample. However, studies in which participants have psychiatric or physical comorbidities will be included but those comorbidities will not be a part of the analysis.

## Types of interventions

We will consider any psychological and psychosocial intervention that is predominantly tailored to treat AH in SSD. Interventions are defined as any psychological or psychosocial intervention that aims to improve AH in SSD, except for music and dance therapy. Two independent reviewers will evaluate this by considering the description of the intervention provided in the studies and, in case of need, by contacting study authors. Since we expect this evaluation will only sometimes be clear, we will also operationalize this decision by looking at the primary outcome of the studies. Interventions must not be developed initially for AH but need to be implemented and adjusted to treat AH as the primary treatment aim. Examples of psychological and psychosocial therapies that are commonly implemented for positive symptoms in SSD are as follows: Cognitive Behavioral Therapy for Psychosis (CBTp), mindfulness-based therapies, acceptance and commitment therapy, family-based interventions, avatar therapy, or digital approaches.

To summarize, we will define the main inclusion criterion for types of interventions as any psychological or psychosocial intervention that predominantly aims to treat AH in SSD, independent of their initial implementation aim. Interventions will be considered as "tailored for AH" when the implemented treatment clearly addresses to ameliorate any characteristics of AH e.g., the frequency, distress, power, control, or beliefs about voices. Additionally, we will include studies with a primary outcome of AH. However, if the primary outcome or treatment aim is unclear or different from AH, the trial will be excluded due to its missing focus on AH. We will deliberately assess the content of the intervention for its usability and efficacy, especially or trans-diagnostically designed to target AH. In case that the intervention is clearly described as addressing AH but analyzes AH as a secondary outcome, it will also be included in the trial. Since previous meta-analysis and systematic reviews already analyzed the efficacy of general psychological interventions for positive symptoms, including AH [e.g., 27], we will not consider studies that examined the overall efficacy of psychological and psychosocial treatments for positive symptoms. There will be no restrictions concerning the implementation of the treatment. Therefore, interventions can be e.g., delivered individually, in a group setting, or digitally. Besides, unguided self-help interventions such as self-help books or online self-help programs will also be allowed. Trials need to be controlled by treatment as usual (TAU), another (non-) pharmacological condition, or a wait-list control group. Additionally, we will allow for non-active interventions such as psychological placebo whereby those need to intend to control for non-specific aspects of the treatment (e.g., befriending, supportive counseling, or social activity therapy). Examples of appropriate controlled designs are as follows:

- psychological/psychosocial intervention *vs.* control (e.g., treatment as usual, waitlist, non-active or active intervention).

- psychological/psychosocial intervention A plus psychological/psychosocial intervention B *vs.* psychological/psychosocial intervention B.

- psychological/psychosocial intervention or psychological/psychosocial intervention plus medication *vs.* medication.

Moreover, interventions with a clear primary treatment aim different from AH will be excluded. Trials with the following interventional aspects will not be considered: (1) Studies focusing on psychopharmacological interventions or trans-magnetic stimulation. (2) Interventions that are predominantly generated to improve other clinical outcomes or symptoms such as overall positive symptoms, delusions, negative symptoms, anxiety, or posttraumatic stress.

## Outcome measures

Outcomes will be measured as defined in each study. However, we will only extract data from studies that used validated rating scales to avoid exaggeration of results [50].

## Primary outcome

Changes in AH (from baseline to post-intervention; or post-intervention) will be the primary outcome. Those include, for example, frequency, distress, duration, loudness, beliefs about origin, and others. Measurements will be any published scale that examines AH, such as the Psychotic Symptom Rating Scale [PSYRATS; 51], the Beliefs about Voices Questionnaire [BAVQ; 52, 53], or others. We will only extract data from studies that used validated rating scales [50].

## Secondary outcomes

The primary aim of the present study is to report the efficacy of targeted psychosocial and psychological interventions on AH in SSD, relying on the effect size extracted from studies. Nevertheless, if reported, we will describe secondary outcomes. We will examine effects on overall symptoms of SSD, negative symptoms, delusions, changes in depressive symptoms, social functioning, or quality of life. Equally, as measurements for the primary outcome, only validated rating scales for secondary outcomes will be considered. We will report drop-out rates as an acceptability outcome of the interventions, defined as the percentage of patient leaving the study prematurely for any reason.

## Information sources and search strategy

We will search for appropriate publications in the following electronic online sources: MEDLINE, PsycINFO, PsycArticles, PSYNDEX, and PubMed. The following search string will be applied to electronic databases: (auditory hallucination* OR hallucination* OR verbal auditory hallucination* OR hearing voice*) AND (psychosis OR psychoses OR schizophrenia OR schizophrenia spectrum disorder* OR psychotic disorder* OR non-affective psychosis) AND (psychological treatment OR psychotherapy OR psychological intervention OR psychosocial intervention OR therapy OR therapies OR treatment). This search string will be adapted if necessary to obtain more specific results. Furthermore, we will browse the Cochrane Library to detect additional literature. For detailed drafts of the search strategy, see S1 Table. We will also search previous reviews and meta-analyses conducted on psychosocial and psychological treatments for AH to check if they included studies that also meet our inclusion criteria.

## Data management and selection process

Studies identified through the described sources will be administered in Rayyan, a web tool for screening and selecting studies in systematic reviews [54]. Duplicates will be removed, and the first basic exclusion criteria will be applied (e.g., meta-analyses and reviews, books, and case reports will be excluded). We will implement a two-stage procedure for identifying eligible studies: First, two independent researchers will screen titles and abstracts. Trials that are not pertinent will be excluded, respectively. Disagreements will be solved by discussion. If the disagreement cannot be resolved and there is still doubt about the eligibility, the entire article will proceed to the next full-text screening stage. Second, the two independent researchers will read the full articles and include those that meet all the applied inclusion criteria. Equally, disagreements will be resolved by discussion. If the doubt about eligibility still remains, a third independent rater will be consulted. In case of persisting uncertainties, we will contact the authors of the original studies by email to provide further information and/or clarification.

The selection process will be presented in a flow chart according to the Preferred Reporting Items for Systematic Reviews and Meta-Analyses [46].

## Data extraction

The first author (L.F.) will extract data from the selected studies. If any uncertainties about the data extraction arise, the co-authors will be consulted, and residual aspects will be discussed. The following data will be collected and reviewed from any included trial, respectively:

1. **Overall study characteristics**, including study citation, year of publication, setting, and sample size.

2. **Methodology**, including study design (type of RCT), study arms, blinding (if outcome assessors are involved), statistical analyses, and risk of bias (see section *Risk of Bias in Individual Studies*).

3. **Characteristics of participants**, including gender, age, diagnosis, sample sizes for each arm, relevant sociodemographic characteristics, and dropouts.

4. **Characteristics of the intervention** include type of intervention, setting (e.g., group vs. individually, outpatient vs. inpatient clinic), number and frequency of sessions, type of therapeutical contact, and type of comparators.

5. **Outcome measures**, including methods of diagnostic assessment and specific measures used, applied time points, the effect of primary outcome measures, and the effect of secondary outcome measures.

## Risk of bias in individual studies

To assess the risk of bias in the studies included, we will use the Risk of Bias [RoB2; 55] Tool recommended by the Cochrane Collaboration (www.riskofbias.info). The RoB2 Tool covers all types of bias that might affect the results of randomized trials. We will assess the risk of bias for each study, respectively. According to the RoB2 Tool, the following bias domains will be assessed:

- Bias arising from the randomization process.

- Bias due to deviations from intended interventions.

- Bias due to missing outcome data.

- Bias in measurement of the outcome.

- Bias in the selection of the reported result.

- Other biases.

Besides, the following three categories will serve as the risk of bias judgment: 'low risk of bias,' 'some concerns', and 'high risk of bias.' In case of insufficient information to make a valid judgment, we will indicate an 'unclear risk of bias.' Effects of high risk of bias will be analyzed by a sensitivity analysis (see section *Sensitivity Analysis*). To additionally assess the potential risk of publication bias in meta-analysis, we will conduct and examine funnel plots according to recommendations [56].

## Data synthesis

**a. Systematic review.** We will produce descriptive statistics and systematically review the characteristics of the included primary studies (for more information about the data, see

Section *Data Collection Process*). Moreover, we will categorize the interventions into separate groups depending on their individual content (e.g., avatar therapy, CBTp, mindfulness interventions).

**b. Meta-analysis.**   Where possible, we will synthesize the extracted data into a meta-analysis. Due to a continuous primary outcome (changes in AH), we will use standardized mean differences (SMD) that are appropriate for post-intervention measurements or change-from-baseline measures [57]. Besides, it suits studies that assess the same outcome but measure it differently, e.g., using different psychometric scales. In general, we will use SMD for continuous outcomes and the risk ratio (RR) for dichotomous outcomes. Both will be presented with 95% confidence intervals (CIs). To address expected heterogeneity across trials, we will use a random-effect meta-analysis. For the illustration of results, we will use forest plots.

**c. Dealing with missing data and missing statistics.**   We will extract data for all randomized participants, preferring data based on intention-to-treat methods if reported. We will contact the original authors in case of missing or unusable data. Calculations of missing values will be carried out by the guidelines of the Cochrane Handbook for Systematic Reviews [57], using imputation methods. We will report Standard Deviations (SDs) where available. If SDs are missing, we will convert the missing data based on other reported estimates, such as SEs, P values, or CIs, following the recommendations in the Cochrane Handbook for Systematic Reviews [57].

**d. Assessment of heterogeneity.**   The heterogeneity across studies will be measured with $I^2$. We will use the following interpretation of heterogeneity classification according to Higgins, Thompson [58]: 0% = no heterogeneity, 25% = low heterogeneity, 50% = moderate heterogeneity, and 75% = high heterogeneity. We will visualize the heterogeneity across studies using funnel plots.

**e. Sensitivity analysis.**   Since many relevant aspects for sensitivity analysis are primarily identified during the reviewing process [57], we will determine suitable issues for the sensitivity analysis during the in-depth examination of the included literature. However, we plan to exclude studies for the sensitivity analysis with a 'high risk of bias' and examine dose-response models regarding the number of sessions implemented or different settings (individual vs. group), noting that this is only a first concept without implementation safety.

## Statistical software

Using suitable packages, statistical analyses will be conducted in R [latest version; 59].

## Ethics and dissemination

Since the proposed study will only consist of existing anonymous data, there is no formal ethical review or assessment required. We will disseminate the study's findings in peer-reviewed journals and through conference presentations. Furthermore, we will publish the results on collaborative platforms or public databases to increase the reach of our findings.

## Discussion

In recent years, a growing body of research on psychological interventions for SSD and AH suggests the efficacy of non-pharmacological approaches [21–24]. Available psychological and psychosocial interventions for positive symptoms in SSD commonly address general treatment effects rather than focusing on specific symptomatic categories. Even though some trials and meta-analyses examined treatment effects on positive symptoms and occasionally on specific symptomatic categories [27], there are no overall efficacy analyses of specifically targeted interventions for AH in SSD. Since the first overviews and analyses suggest higher efficacy of

specialized treatments based on CBTp compared to general approaches [44, 45], future research needs to evaluate those outcomes further.

Due to the still existing shortcomings in the knowledge of targeted treatments for AH, the present review and meta-analysis will provide insights into an evidence synthesis of the efficacy of any psychological and psychosocial interventions targeted at treating AH in SSD. One of the critical methodological strengths of this study is the restriction to RCTs that will lead to decreased sources of bias that are common in trials without control comparators. Besides, this study will include various forms of interventions that comply with settings in the clinical routine. However, the following methodological limitations must be considered. Since we will only include RCTs published in English or German, we might lose some data close to the real-life context of clinical procedures. Moreover, findings will be limited by publication bias, the measurements used in each trial, study heterogeneity, and the methodological quality of the included trials. Nevertheless, we will address those limitations with the RoB2 Tool [55].

To the best of our knowledge, this will be the first systematic review and meta-analysis that intends to inform about all available psychological and psychosocial interventions especially tailored to treat AH in persons with SSD. Examining specific treatment effects will augment the existing evidence by providing an overview of convenient treatment approaches and their overall efficacy at treating AH in SSD. Those findings complement existing evidence that may impact future implementation of psychological and psychosocial approaches in clinical routine and, therefore, improve treatment outcomes for the addressed population. Furthermore, the results of the proposed study could influence future guidelines on psychological treatments of AH in SDD to further strengthen clinical care for the addressed population. Due to the predetermined methodological approach in accordance with current recommendations [46], this systematic review and meta-analysis promises to give valid insights into a topic specifically relevant to clinical research and practice.

## Supporting information

**S1 Checklist. PRISMA-P (Preferred Reporting Items for Systematic review and Meta-Analysis Protocols) 2015 checklist: Recommended items to address in a systematic review protocol.**
(DOCX)

**S1 Table. Draft of search strategies that will be applied to data bases, examples.**
(DOCX)

## Acknowledgments

The authors would like to thank Steffen Moritz for his expertise and ciritial feedback on the manuscript of the study protocol.

## Author Contributions

**Conceptualization:** Laura Fässler, Kerem Böge.

**Investigation:** Laura Fässler.

**Methodology:** Laura Fässler, Irene Bighelli, Stefan Leucht, Michel Sabé, Malek Bajbouj, Christine Knaevelsrud, Kerem Böge.

**Project administration:** Laura Fässler.

**Supervision:** Kerem Böge.

**Writing – original draft:** Laura Fässler.

**Writing – review & editing:** Irene Bighelli, Stefan Leucht, Michel Sabé, Malek Bajbouj, Christine Knaevelsrud, Kerem Böge.

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
