## [Decision Letter · Decision Letter 0]

28 May 2024

PONE-D-24-15965Targeted Psychological and Psychosocial Interventions for Auditory Hallucinations in Persons with Psychotic Disorders:

Protocol for a Systematic Review and Meta-AnalysisPLOS ONE

Dear Dr. Fässler,

Thank you for submitting your manuscript to PLOS ONE. After careful consideration, we feel that it has merit but does not fully meet PLOS ONE’s publication criteria as it currently stands. Therefore, we invite you to submit a revised version of the manuscript that addresses the points raised during the review process.

We look forward to receiving your revised manuscript.

Kind regards,

Carmen Concerto

Academic Editor

PLOS ONE

Journal Requirements:

4. We note that this manuscript is a systematic review or meta-analysis; our author guidelines therefore require that you use PRISMA guidance to help improve reporting quality of this type of study. Please upload copies of the completed PRISMA checklist as Supporting Information with a file name “PRISMA checklist”.

Reviewers' comments:

Reviewer's Responses to Questions

**Comments to the Author**

1. Does the manuscript provide a valid rationale for the proposed study, with clearly identified and justified research questions?

Reviewer #1: Yes

Reviewer #2: Yes

2. Is the protocol technically sound and planned in a manner that will lead to a meaningful outcome and allow testing the stated hypotheses?

Reviewer #1: Yes

Reviewer #2: Yes

3. Is the methodology feasible and described in sufficient detail to allow the work to be replicable?

Reviewer #1: Yes

Reviewer #2: Yes

4. Have the authors described where all data underlying the findings will be made available when the study is complete?

Reviewer #1: Yes

Reviewer #2: Yes

5. Is the manuscript presented in an intelligible fashion and written in standard English?

Reviewer #1: Yes

Reviewer #2: Yes

6. Review Comments to the Author

You may also provide optional suggestions and comments to authors that they might find helpful in planning their study.

Reviewer #1: I have reviewed with pleasure the manuscript “Targeted Psychological and Psychosocial Interventions for Auditory Hallucinations in Persons with Psychotic Disorders: Protocol for a Systematic Review and Meta-Analysis” (PONE-D-24-15965) and found it to be well structured and relevant. The manuscript is a valuable contribution to the field of psychological interventions for psychotic disorders. However, minor revisions are suggested to improve clarity and ensure comprehensive topic coverage.

Specifically, the authors should consider the following recommendations:

1. The abstract is concise and informative. However, it would be beneficial to explicitly mention the anticipated impact of this review on clinical practice.

2. The introduction effectively outlines the rationale and need for this review. I suggest briefly mentioning any preliminary findings or pilot data that led to the development of this protocol to provide additional context.

3. The criteria for inclusion of participants with SSD and current AH are clear. However, the Authors should consider specifying how “current AH” is operationally defined across studies to ensure consistency.

4. The exclusion of studies with fewer than 10 participants per arm is reasonable. Still, it may be helpful to justify this threshold based on the potential impact on statistical power or variability.

5. While the focus on tailored interventions is commendable, defining “tailored” more explicitly would be helpful. For example, what specific characteristics or components qualify an intervention as “tailored” to AH?

6. The plan for dissemination of results through peer-reviewed journals and conferences is appropriate. Consider mentioning additional dissemination strategies, such as public databases or collaborative platforms, to increase the reach and impact of the results.

Reviewer #2: The manuscript presents a well-structured protocol for systematic review and meta-analysis aimed at assessing the effectiveness of targeted psychological and psychosocial interventions for auditory hallucinations in schizophrenia spectrum disorders. Adhering to PRISMA guidelines, it outlines a comprehensive methodology, and is set to fill a notable void in existing research on non-pharmacological treatments for auditory hallucinations. The paper warrants acceptance, with a few minor considerations for improvement.

- The title might be more informative if it explicitly referenced "non-pharmacological" interventions, thereby clarifying the scope of the review to potential readers.

- The theoretical underpinnings of interventions such as CBTp and ACT could be more thoroughly examined, discussing on the mechanisms through which these treatments exert their effects.

- It may be worthwhile to highlight how personalized interventions could potentially improve both adherence to treatment and patient outcomes, emphasizing the review's relevance.

- The approach to comorbid conditions, such as depression and anxiety, should be clarified. Will these be accounted for within the primary analyses, or will they be the subject of separate subgroup analyses?

- The inclusion of additional statistical methods, such as Egger's test, could help to assess publication bias, thereby strengthening the meta-analysis.

- Placing the anticipated effect sizes in the context of those reported in broader meta-analyses on the topic (non-pharmacological intervention for schizophrenia and pharmacological types of intervention for AH) could provide valuable perspective on the expected findings.

- Finally, a discussion on the potential clinical implications of the review's findings would be beneficial, such as how they might influence treatment guidelines or inform clinical practice.

Best Regards.

7. PLOS authors have the option to publish the peer review history of their article (what does this mean?). If published, this will include your full peer review and any attached files.

Reviewer #1: No

Reviewer #2: **Yes: **Pierfelice Cutrufelli

---

## [Author Response · Author response to Decision Letter 0]

12 Jun 2024

Journal Requirements:

Thank you for this feedback! We updated the manuscript and the file names and hope to now fulfil all style requirements of PLOS ONE.

We will ensure this aspect when resubmitting the documents. Since the awards are not received especially for this study but rather as a personal scholarship / funding, we cannot provide a grant number. However, we will include written confirmation of the grand providers for clarification (please do not publish those confirmations). Additionally, we included the funding disclosure in the cover letter.

Thank you for this reminder! We included captions for our supporting information files at the end of our manuscript, including the edited file names.

4. We note that this manuscript is a systematic review or meta-analysis; our author guidelines therefore require that you use PRISMA guidance to help improve reporting quality of this type of study. Please upload copies of the completed PRISMA checklist as Supporting Information with a file name “PRISMA checklist”.

Since we named the PRISMA checklist “Supporting information_Table S1”, it may have been overlooked. We updated the file name entitled “PRISMA checklist” and hope it suits your guidelines. 

Thank you for mentioning this very important aspect! We searched for every included reference in our study to be complete and correct. Additionally, we searched MEDLINE / PubMed with the terms “schizophrenia” and “auditory hallucinations” for retracted articles that might be included in our reference list. We could not find any article that was retracted. We hope to address your remarks appropriately. If there should be any retracted articles included in our reference list, we will, of course, remove them immediately. Please contact us if we overlooked anything!

Reviewer #1:

I have reviewed with pleasure the manuscript “Targeted Psychological and Psychosocial Inter-ventions for Auditory Hallucinations in Persons with Psychotic Disorders: Protocol for a Systematic Review and Meta-Analysis” (PONE-D-24-15965) and found it to be well structured and relevant. The manuscript is a valuable contribution to the field of psychological interven-tions for psychotic disorders. However, minor revisions are suggested to improve clarity and ensure comprehensive topic coverage. Specifically, the authors should consider the following recommendations:

First, thank you very much for your kind words – we are very pleased about your positive feedback. Furthermore, we would like to thank you for your constructive remarks, which helped us to improve our study design and the overall quality of our manuscript. We will systematically address the suggested aspects in the following sections.

1. The abstract is concise and informative. However, it would be beneficial to explicitly mention the anticipated impact of this review on clinical practice.

Thank you for this important suggestion. We included one sentence in the abstract to explicitly address the impact of the review on clinical practice that should clarify the anticipated implications (page 2, line 48).

Page 2, line 48: ‘These findings will complement existing evidence that may impact future treatment implementations in clinical practice by addressing effective strategies to treat AH and therefore improve outcomes for the addressed population.’

2. The introduction effectively outlines the rationale and need for this review. I suggest briefly mentioning any preliminary findings or pilot data that led to the development of this protocol to provide additional context.

Again, thank you for this helpful comment. We have now outlined the results of the pi-lot and previous data on tailored approaches to auditory hallucinations (AH) in more detail to provide a reasonable context for the development of our research idea and protocol. We hope that this addition complies with your suggestion.

Page 6, lines 130-135: ‘Previous research on treatment effects of tailored approaches addressing AH found large effect sizes for experienced focused counselling (d = 1) [43], moderate effects for a reduction in the negative impact of voices for individual mindfulness-based interventions [37], or significant reduction in harmful compliance with and an increase in the perceived control over voices with small-to-moderate effect sizes [31].’

3. The criteria for inclusion of participants with SSD and current AH are clear. However, the Authors should consider specifying how “current AH” is operationally defined across studies to ensure consistency.

We appreciate this critical remark as well! Therefore, we added a specification of “currently occurring AH” on page 9, lines 210-214. These studies need to explicitly mention current AH as an inclusion criterion and requirement for study participation. This will be operationalized over rating scales, self-reports, screening procedures, or clinical documents. 

Page 9, lines 210-214: ‘However, participants must have current AH, explicitly mentioned as an inclusion criterion or requirement for participation. This needs to be operationalized and ensured either by ratings scales measuring the frequency of AH with a predefined cut-off score, over outcome measures, self-reports, screening procedures, or clinical documents such as physician’s letters.’

4. The exclusion of studies with fewer than 10 participants per arm is reasonable. Still, it may be helpful to justify this threshold based on the potential impact on statistical power or variability.

Again, this is a thoughtful remark that we considered in our revision. To clarify the rea-sons for the sample size restriction as an exclusion criterion, we outlined the potential impacts of a sample size <10 per study arm on validity, sensitivity of treatment effects, and statistical power with relevant literature.

Page 8, lines 190-192: ‘Case series and trials with less than 10 participants in each study arm will be excluded to ensure sensitivity for treatment effects [47], validity, and statistical power of results [48].’

5. While the focus on tailored interventions is commendable, defining “tailored” more explicitly would be helpful. For example, what specific characteristics or components qualify an inter-vention as “tailored” to AH?

This is a very important remark which we highly appreciate. We addressed this feed-back on page 10, lines 239 to 241, by defining interventions “tailored for AH” when they explicitly address and were developed to ameliorate characteristics of AH, such as frequency, distress, or control over voices. Thank you for this helpful comment!

Page 10, lines 239-241: ‘Interventions will be considered as “tailored for AH” when the implemented treatment clearly addresses to ameliorate any characteristics of AH e.g., the frequency, distress, power, control, or beliefs about voices.’

6. The plan for dissemination of results through peer-reviewed journals and conferences is appropriate. Consider mentioning additional dissemination strategies, such as public data-bases or collaborative platforms, to increase the reach and impact of the results.

Thank you again for this remark. We considered your suggestion on page 17, lines 393 & 394. 

Page 17, lines 393 & 394: ‘Furthermore, we will publish the results on collaborative platforms or public databases to increase the reach of our findings.’

Reviewer #2:

The manuscript presents a well-structured protocol for systematic review and meta-analysis aimed at assessing the effectiveness of targeted psychological and psychosocial interventions for auditory hallucinations in schizophrenia spectrum disorders. Adhering to PRISMA guide-lines, it outlines a comprehensive methodology, and is set to fill a notable void in existing re-search on non-pharmacological treatments for auditory hallucinations. The paper warrants acceptance, with a few minor considerations for improvement.

We thank you very much for your time in reviewing our study protocol and your positive feedback, as well as the critical considerations that supported us in improving the article. We highly appreciate every comment that will be addressed in the following, respectively. 

- The title might be more informative if it explicitly referenced "non-pharmacological" interven-tions, thereby clarifying the scope of the review to potential readers.

We totally understand your point – thank you! Since we mentioned in the title “Targeted psychological and psychosocial interventions for auditory hallucinations” we wanted to clarify our focus on non-pharmacological interventions. Therefore, the title might get too long if we would additionally include the specific term “non-pharmacological”. Furthermore, we will focus on psychological and psychosocial interventions rather than on the whole spectrum of non-pharmacological approaches which would include additional treatments. We hope our thoughts are comprehensible.

- The theoretical underpinnings of interventions such as CBTp and ACT could be more thoroughly examined, discussing on the mechanisms through which these treatments exert their effects.

We added some aspects of the current evidence concerning the mechanisms of action regarding various psychological and psychosocial approaches especially targeted to treat AH in SSD on page 5, lines 115-121. Thank you for this thoughtful feedback!

Page 5, lines 115-121: ‘Despite this, approaches such as relating therapy [35] or mindfulness-based interventions for voice hearing [36, 37] are implemented to address distressing AH. Targeted psychological interventions for AH emerging from CBTp focus on an empathetic and accepting perspective to change the persons’s relationship with the voices rather than their content or quantity [38-40]. Accordingly, affected persons can increase their perceived control over the appearance of the voices and, therefore, decrease the associated dis-tress [38].’

- It may be worthwhile to highlight how personalized interventions could potentially improve both adherence to treatment and patient outcomes, emphasizing the review's relevance.

This is also an important point we highly appreciate. To address this suggestion, we highlighted the potential amelioration of treatment adherence on page 5, lines 107-109.

Page 5: lines 107-109: ‘Besides, they can address an amelioration of treatment adherence for psychological approaches by its individual formulation [e.g., 29] This is especially relevant due to the high non-adherence rates in SSD, mainly for pharmacological interventions [30].’

- The approach to comorbid conditions, such as depression and anxiety, should be clarified. Will these be accounted for within the primary analyses, or will they be the subject of separate subgroup analyses?

Thank you for your thoughts on that topic! Comorbid conditions will be allowed in the chosen study population. However, studies that analyze AH in another primary diagnosis sample than schizophrenia spectrum disorders (SSD) will be excluded. Comorbidities will not be part of the meta-analysis, as additionally, we clarified on page 10, line 223. Despite this, further reported symptoms such as overall symptoms of SSD, negative symptoms, delusions, changes in depressive symptoms, social functioning, or quality of life will be examined as secondary outcomes. You can find this on page 7, lines 162-163, as well as on page 12, lines 282-285. We hope this will clarify our approach toss secondary outcomes and comorbidities. 

Page 10, line 223: ‘However, studies in which participants have psychiatric or physical comorbidities will be included but those comorbidities will not be a part of the analysis.’

- The inclusion of additional statistical methods, such as Egger's test, could help to assess publication bias, thereby strengthening the meta-analysis.

Thank you for this very important aspect! We searched for adequate statistical methods to assess possible publication bias arising from the included studies. To follow recommenda-tions in conducting meta-analyses (Sterne et al., 2011), we will implement funnel plots that will visualize potential publication bias. You can find this addition on page 15, lines 350-352. Thank you again; this will help strengthen the methods of the present meta-analysis.

Page 15, lines 350-352: ‘To additionally assess the potential risk of publication bias in meta-analysis, we will conduct and examine funnel plots according to recommendations [56].’

- Placing the anticipated effect sizes in the context of those reported in broader meta-analyses on the topic (non-pharmacological intervention for schizophrenia and pharmacological types of intervention for AH) could provide valuable perspective on the expected findings.

According to the comment of reviewer #1, who also suggested reporting preliminary ef-fect sizes of findings from studies examining the effects of tailored approaches for AH, we outlined the results of pilot and previous data, including effect sizes (page 6, lines 130-135) as well as a previous meta-analysis on tailored CBTp approaches for AH (page 6, line 135-137). Fur-thermore, we mentioned the effect sizes of meta-analysis analyzing non-pharmacological interventions for positive symptoms in SSD (page 5, lines 100-103). To outline those findings in the context of our expected results, we added one more aspect on pages 7 & 8, lines 167-169. We hope to have addressed your feedback accordingly! 

Page 7 & 8, lines 167-169: ‘According to previous findings on different psychological and psychosocial interventions for AH and SSD [27, 31, 44, 45], we expect moderate effect sizes for tailored approaches for AH.’

- Finally, a discussion on the potential clinical implications of the review's findings would be beneficial, such as how they might influence treatment guidelines or inform clinical practice.

Thank you for your helpful comment. We added some aspects in the discussion to fur-ther outline the potential clinical implications (page 18, lines 421-426).

Page 18, lines 421-426: ‘Those findings complement existing evidence that may impact future implementation of psychological and psychosocial approaches in clinical routine and, therefore, improve treatment outcomes for the addressed population. Furthermore, the results of the proposed study could influence future guidelines on psychological treatments of AH in SDD to further strengthen clinical care for the addresses population.’

Overall, we would like to thank the editors and reviewers for their constructive and helpful feedback, which supported us in increasing the quality of the current study protocol and its methods.

You can also find our response to the reviewers in a separate file ("Response to Reviewers").

---

## [Editor Report · Decision Letter 1]

16 Jun 2024

Targeted Psychological and Psychosocial Interventions for Auditory Hallucinations in Persons with Psychotic Disorders:

Protocol for a Systematic Review and Meta-Analysis

PONE-D-24-15965R1

Dear Dr. Fässler,

We’re pleased to inform you that your manuscript has been judged scientifically suitable for publication and will be formally accepted for publication once it meets all outstanding technical requirements.

Kind regards,

Carmen Concerto

Academic Editor

PLOS ONE
---

## [Editor Report · Acceptance letter]

24 Jun 2024

PONE-D-24-15965R1 

PLOS ONE

Dear Dr. Fässler, 

I'm pleased to inform you that your manuscript has been deemed suitable for publication in PLOS ONE. Congratulations! Your manuscript is now being handed over to our production team.

Kind regards, 

on behalf of

Dr. Carmen Concerto 

Academic Editor

PLOS ONE